

# Influence of leaf inclination angle and tillering on population transpiration, soil evaporation, and yield in winter wheat near-isogenic lines

Guirong Huang[1,2], Xinying Zhang[1,3], Zhenzhao Wang[1], Xiaoying Liu[1], Rui Guo[1], Fengxue Gu[1], Enke Liu[1], Shuying Li[4], Xiuli Zhong[1], Qiaozhen Li[1] and Xurong Mei[1]

[1] Key Laboratory for Dryland Agriculture of Ministry of Agriculture and Rural affairs, Institute of Environment and Sustainable Development in Agriculture, Chinese Academy of Agricultural Sciences, Beijing, The People's Republic of China
[2] Agriculture and Rural Bureau of Jingxing District, Shijiazhuang, The People's Republic of China
[3] Hebei Key Laboratory of Crop Stress Biology, College of Agronomy and Biotechnology, Hebei Normal University of Science & Technology, Qinhuangdao, The People's Republic of China
[4] Forestry and Grassland Bureau of Aohan Banner, Chifeng, The People's Republic of China

Corresponding authors
Qiaozhen Li, liqiaozhen@caas.cn
Xurong Mei, meixurong@caas.cn

## ABSTRACT

Leaf inclination angle (LIA) and tillering impact the winter wheat (*Triticum aestivum* L.) population canopy structure. Understanding their effects on water use (WU) parameters and yield can guide water-saving strategies through population control. In this study, six near-isogenic lines (NILs) and their parents were selected as materials. These special materials were characterized by varying tillering at the current sowing density, a similar genetic background, and, particularly, a gradient in mean flag leaf LIA. The investigation focused on the jointing to early grain-filling stage, the peak water requirement period of wheat crops. Population-scale transpiration (PT) and evaporation from the soil surface (E) were partitioned from total evapotranspiration (ET) by the means of micro-lysimeters. The results showed decreased PT, E, and ET with increased population density (PD) within a narrow density range derived from varying tillering across genotypes. Significant correlations existed between PD and ET, E, and PT, especially in the wettest 2017–2018 growing season. Within such narrow PD range, all the correlations between WU parameters and PD were negative, although some correlations were not statistically significant, thereby suggesting the population structure's predominant impact. No significant correlation existed between LIA and both ET and PT within the LIA range of 35°–65°. However, significant correlations occurred between LIA and E in two growing seasons. Genotypes with similar LIA but different PD produced varied ET; while with similar PD, the four pairs of genotypes with different LIA each consumed similar ET, thus highlighting PD's more crucial role in regulating ET. The yield increased with higher LIA, and showed a significant correlation, emphasizing the LIA's significant effect on yield. However, no correlation was observed with PD, indicating the minor effect of tillering at the current sowing density. Therefore these results might offer valuable insights for breeding water-saving cultivars and optimizing population structures for effective field water conservation.

## INTRODUCTION

The North China Plain is one of the most important agricultural regions in China. The rich soil and favorable climate are suitable for growing winter wheat (*Triticum aestivum* L.) and summer maize (*Zea mays* L.) as the most common double-cropping system. The typical continental monsoon climate dominates the region, leading to an extremely variable temporal distribution of annual rainfall. Over 70% of annual precipitation falls during the maize-growing season (from July to September), with <30% occurring in the wheat-growing season. This meets only about 25–40% of the water requirements of wheat, necessitating irrigation for high wheat production. Hence, over 70% of the irrigation water is used for winter wheat (*Sun et al., 2006*; *Zhang, Qin & Xie, 2016*). The region faces a critical water situation due to excessive exploitation of groundwater for irrigation from both shallow and deep aquifers, leading to a rapid decline in the groundwater table in some areas (*Yang et al., 2015*). Furthermore, projections indicate an impending increase in water demand and exacerbation of the existing water deficit in the North China Plain over the next two decades (*Cao, Han & Song, 2014*). Efficient pathways to reduce water use while maintaining the current high wheat production levels in the North China Plain urgently need to be explored for ensuring national food security and mitigating regional water resource crisis.

Selecting and breeding water-saving cultivars offers a long-lasting and cost-effective approach to water conservation. Breeding efforts in the North China Plain over the past several decades focused on enhancing yield in irrigated areas and improving drought tolerance to gain good and stable harvests in rain-fed areas (*Yang et al., 2015*; *Zhang et al., 2017*), but overlooked reducing water use (WU). Actually, the genotypic variation in water use efficiency is affected more by WU variation, the denominator, than by biomass variation, the nominator (*Blum, 2005*). This emphasizes the potential for significant water conservation by breeding cultivars with lower water consumption, complemented by appropriate cultivation practices.

The field population's WU differs from the transpiration of a single leaf or individual plant. It primarily comprises population-scale transpiration (PT) and evaporation from the soil surface (E), with PT representing over 70% of the total evapotranspiration (ET) in the North China Plain (*Wang et al., 2001*). Soil evaporation, a physical process involving water molecule diffusion at the soil-air interface, is influenced by environmental factors beyond the canopy and microclimate within it (*Eberbach & Pala, 2005*). Transpiration itself, a physical process occurring at the leaf-air interface, involves water molecules freely diffusing into the air through leaf stomata (*Jarvis & Mcnaughton, 1986*). Crops regulate stomatal aperture to control transpiration, while meeting $CO_2$ uptake requirement for photosynthesis (*Hetherington & Woodward, 2003*). On the other hand, transpiration is strongly influenced by surrounding environmental factors (*Nguyen et al., 2022*). Given the similar stomatal apertures, the genotypic difference in PT depends on the microclimate

within the canopy, which is determined by canopy structure (*Gao et al., 2002*). Thus, both PT and E are regulated by canopy structure, which forms a canopy resistance to limit PT and E (*Holmes & Keiller, 2002*). Crop canopy structure is predominantly determined by two major features: plant density and plant architecture (*Dornbusch et al., 2011*; *Huang et al., 2017*; *Li et al., 2021*). For cereal crops that possess the ability of tillering, the population consists of all shoots, including main stem, fertile tillers, and infertile tillers. Differing from sowing/planting density, population density is thus defined as shoot density (SD) , commonly designated as shoots m$^{-2}$. Plant architecture is a specific three-dimensional architecture jointly determined by several main attributes like leaf inclination angle (LIA), leaf size and shape (*Tafesse Endale, Warkentin Thomas & Bueckert Rosalind, 2019*). LIA affects foliage arrangement in a population, influencing light interception and radiation use efficiency (*Hikosaka & Hirose, 1997*; *Toler et al., 1999*; *O'Toole, Ito & Hardy, 1998*). Previous studies examined the effect of LIA on key physiological processes such as photosynthesis and yield (*Jost et al., 2006*; *Kaggwa-Asiimwe, Andrade-Sanchez & Wang, 2013*). Studies found that upright leaves facilitate deeper light penetrate and more uniform distribution within the canopy, thus enhancing carbon assimilates accumulation (*Wullschleger & Oosterhuis, 1990*; *Marois et al., 2004*). *Falster & Westoby (2003)* found shallow-angled leaves had greater daily light interception and potentially greater carbon gain than steeper-angled leaves. Modifying plant architecture traits is also considered a critical breeding strategy to enhance crop yield potential (*Khush, 2005*). However, the impact of LIA on WU, especially on PT, remains unclear. This study aimed to elucidate how the plant architecture trait LIA influences WU parameters.

SD significantly impacts population canopy structure. A number of studies (*Geleta et al., 2002*; *Tollenaar et al., 2006*) have explored optimal sowing density effects on grain yield, while others focused on its impact on SD in winter wheat (*Bulson, Snaydon & Stopes, 1997*; *Spink et al., 2000*). Genetic yield potential improvement is linked to adaptation to agronomic planting density (*Reynolds et al., 1994*; *Sukumaran et al., 2015*). Enhancing plant resilience to high-density planting significantly boosts yield per hectare in wheat and other intensively cultivated crops (*Whaley et al., 2000*). Additionally, cereal crops, including wheat, exhibit tillering as a crucial self-adjustment trait (*Spink et al., 2000*). SD in cereal crops involves total shoots per area unit, including main shoots and tillers. Thus, tillering is another influencing factor for SD besides sowing density. Wheat crops sowed at a lower sowing density can increase shoot number to gain higher SD through tillering. However, the association between tillering, a SD-influencing factor, with WU parameters has been rarely reported.

Larger LIA and higher SD reduce bare soil exposure and minimize evaporation, thereby conserving more water. Once wheat crops reach the jointing stage, the complete canopy amplifies the impact of canopy structure on WU, especially on PT. Thus, focusing on the period from jointing to early grain-filling stages, the peak water requirement phase for wheat crops, the present study aims to (1) elucidate the influence of LIA and tillering on WU traits, especially ET, E, and PT, and (2) establish the associations of LIA and tillering with WU parameters E, PT, and ET, along with grain yield. The findings might offer

valuable insights for breeding water-saving cultivars based on plant architecture traits and optimizing population structure for effective field water conservation.

## MATERIALS AND METHODS

### Experimental site

The experiments were conducted at the Shunyi Scientific Experimental Station, which is located in the Beijing area, the northern part of the North China Plain, affiliated with the Institute of Environment and Sustainable Development in Agriculture, Chinese Academy of Agricultural Sciences (CAAS), in the 2015–2016, 2016–2017, and 2017–2018 growing seasons of winter wheat. The experimental field's soil was classified as calcareous fluvo-aquic (sand 28.7%, silt 64.2%, and clay 7.1%), exhibited a soil bulk density of 1.40 g cm$^{-3}$, and a pH of approximately 7.7. The soil nutrient levels were as follows: 0.109 g/kg of total nitrogen, 14.4 g/kg of organic matter, 24.5 mg/kg of available phosphorus, and 106 mg/kg of available potassium. The previously planted crop was field peas (*Pisum sativum* spp. *arvense*).

Table 1 outlines the monthly mean air temperature and precipitation for the three experimental growing seasons. The highest total precipitation occurred in 2015–2016 (173.4 mm), followed by 2017–2018 (130.1 mm), with 2016–2017 being the driest at 119.6 mm. April, the largest water requirement period of wheat crops in the North China Plain, witnessed meager precipitation in April 2016 (5.6 mm) and 2017 (0.1 mm), but April 2018 recorded 52.7 mm, surpassing long-term average (20.6 mm) by 32.1 mm. The monthly mean air temperatures were relatively higher in 2016–2017, lower in 2017–2018, and moderate in 2015–2016. Across the three years, temperatures were consistently higher from March to June but notably lower in November compared to the long-term average.

### Experimental design

Winter wheat cultivars Jinmai 47 and Jing 411, used as the recurrent and donor parents, respectively, along with their six near-isogenic lines (NILs) 908092, 908120, 908188, 908206, 908032, and 908274, were chosen as experimental materials. They were obtained from the Institute of Crop Sciences, CAAS. Jinmai 47 is a loose, flat-leafed type with an LIA of 65°, and Jing 411 is a compact, upright-leafed type with an LIA of 35°, both of which were widely planted in the North China Plain. These parents and their six NILs were selected based on LIA, forming a gradient from 35° in Jing 411 to 65° in Jinmai 47.

The eight genotypes were planted on October 5, 2015, October 3, 2016, and October 3, 2017, and were harvested on June 12, 2016, June 11, 2017, and June 14, 2018, respectively. The seeding rate was 675 seeds m$^{-2}$, which is the current conventional and widely adopted sowing density in the target region. Two water treatments were implemented: rain-fed and supplemental irrigation treatments. A 2.5-m-wide isolation belt separated the two water treatments, with winter wheat growing in the 1.5-m-wide middle area. Before sowing, all treatments received 90-mm irrigation was fulfilled for all treatments to ensure uniform soil water conditions and seedling emergence. For the rain-fed treatment, no irrigation was applied during the winter wheat-growing season. For the supplemental irrigation treatment, each plot received 120-mm irrigation *via* sprinkling water, 60 mm each before
**Table 1    Monthly precipitation and air temperature for winter wheat growing seasons from 2015 to 2018.**

|  | Growing seasons | Oct. | Nov. | Dec. | Jan. | Feb. | Mar. | Apr. | May. | Jun. | Total |
|---|---|---|---|---|---|---|---|---|---|---|---|
| Rainfall (mm) | 2015-2016 | 8.7 | 42.9 | 2.3 | 1.2 | 9.2 | 0 | 5.6 | 56.9 | 46.6 | 173.4 |
|  | 2016–2017 | 70.8 | 6.5 | 0 | 0.8 | 3.7 | 11.6 | 0.1 | 23.1 | 3 | 119.6 |
|  | 2017–2018 | 58.8 | 0 | 0 | 0 | 0 | 1.7 | 52.7 | 13.5 | 3.4 | 130.1 |
| Mean air temperature (°C) | 2015–2016 | 13.7 | 3 | −0.6 | −4.8 | 0.6 | 8.7 | 16.4 | 21.2 | 25.2 | _ |
|  | 2016–2017 | 13 | 4 | −0.4 | -2 | 1.7 | 8.4 | 17.2 | 23.2 | 25.6 | _ |
|  | 2017–2018 | 12.7 | 3.4 | −1.4 | −3.9 | −1.6 | 8.1 | 15.6 | 22.1 | 26.8 | _ |

Notes.
   Precipitation in June for three growing seasons was sum from June 1 to harvest day.

jointing and before flowering stages. The eight materials were arranged in a randomized block design with three replicates in each water treatment area, and each plot measured $10.0 \times 2.4 \, \text{m}^2$.

## MEASUREMENTS

### LIA and SD
The LIA, representing the angle between the vertical and the leaf plane extension, was measured. The preliminary measurements indicated similarity in the flag leaf LIA between main shoots and tillers in one wheat plant. Hence, LIA measurements were conducted on ten main shoot flag leaves in the central five rows of each plot using a transparent plastic protractor between 10:00 and 11:00 a.m. on windless days at anthesis stage. SD investigation involved manually counting shoots per $\text{m}^2$ in the central five rows at anthesis stage.

### WU traits
Total WU or ET, E, and PT were determined for the period from jointing to early grain filling stage, representing the peak water requirement period for wheat crops, in 2016, 2017, and 2018.
   ET was calculated from the initial soil water content minus final soil water content, precipitation, irrigation, runoff, and drainage using the following equation (*Li et al., 2012*):

$$ET = I + P - R - D - SW$$

where ET is evapotranspiration (mm) from jointing to early grain filling stages; I is irrigation (mm); P is precipitation (mm); R is the surface runoff (mm); and D is downward drainage (mm). R and D were assumed to be 0 due to the low precipitation and deep percolation under the experimental conditions. SW denotes changes in water storage for the 1.6-m soil profile. SW was determined as previously described by *Huang et al. (2023)*. Specifically, the gravimetric soil water contents were measured at 20-cm intervals to a depth of 160 cm by taking soil cores, and then converted into volumetric soil water contents using bulk density data. To mitigate the influence of soil spatial heterogeneity, three sampling points between two rows were randomly arranged in each plot, and samples were taken and fully mixed to obtain a representative sample for each soil layer in each plot. Measurements for
gravimetric soil water contents were conducted on April 17 and May 11 in 2016, April 18 and May 12 in 2017, and April 20 and May 11 in 2018.

The E was determined by the micro-lysimeter method as previously described in *Huang et al. (2023)*. Specifically, the micro-lysimeters were made of polyvinyl chloride cylinders (PVC) (15 cm length and 10 cm inner diameter), and slightly larger PVC cylinders with 12 cm inner diameter and the same length with the instrument serving as the sleeves to facilitate insertion and removal of the micro-lysimeters. The sleeve was pushed into the ground, and emptied of soil prior to installing the micro-lysimeter. Subsequently, the micro-lysimeter was pushed into the soil between two rows within the same plot, carefully taken out, and the soil outside it was cleaned, with the bottom being sealed with water-proof adhesive tape, and then weighed and put into the sleeve. Each plot had three micro-lysimeters installed between rows. For water loss determination, the soil inside the micro-lysimeter was weighed daily at dusk using an electronic balance. The soil inside the micro-lysimeter was changed every three days, and immediately after rainfall or irrigation, to maintain similar soil water contents to the surrounding soil. Soil evaporation was assumed negligible on rainy days. Finally, soil evaporation was calculated based on the daily weight change and the micro-lysimeters' surface area.

PT was finally calculated by subtracting E from ET.

### Yield

Following winter wheat maturation, a 1-m section from the center of each plot, spanning five rows, was randomly hand-harvested. The harvested crop was air-dried and then the grain yield (kg ha$^{-1}$) was calculated.

### Statistical analysis

Statistical analysis was performed using SAS procedure (Version 9.2, SAS Institute, Cary, NC, USA). The results for rain-fed and supplemental irrigation treatments were evaluated separately, because the two water treatments were not randomized within the same study, instead, they were separated by a 2.5-m-wide isolation belt, thus constituting distinct experiments. Additionally, the results for each growing season were analyzed separately, because differences in weather data existed between years, also because the initial combined data showed genotype by year interaction (*Dong et al., 2006*; *Steel & Torrie, 1980*; *Dong et al., 2010*). Based on these, a one-way ANOVA (analysis of variance) was carried out in experimental indices (SD, three WU traits, and yield) under each water condition per growing season to primarily evaluate the differences across eight genotypes. The means of the eight genotypes were compared using the least significant differences at a 0.05 probability level. The regression analyses between experimental indices and WU traits, as well as yield were conducted using Pearson correlation analysis and single regression line.

## RESULTS

### LIA and tillering of different wheat genotypes

In considering flag leaf LIA, the recurrent (Jinmai 47) and donor (Jing 411) parents, along with their six NILs, were categorized into three groups. Group I, comprising Jing 411 and

**Table 2  Flag leaf inclination angles (LIA) of different wheat genotypes.**

|  | Group I | | | Group II | | | Group III | |
| --- | --- | --- | --- | --- | --- | --- | --- | --- |
| Genotypes | Jing 411 | 908274 | 908092 | 908206 | 908120 | 908032 | 908188 | Jinmai 47 |
| LIA (°) | 35 | 39 | 41 | 49 | 52 | 60 | 62 | 65 |

**Notes.**
The flag leaf inclination angles were measured under rain-fed condition in 2017.

NILs 908274 and 908092, exhibited smaller LIA. In contrast, Group III, including Jinmai 47 and NILs 908188 and 908032, had larger LIA. Group II, with medium LIA, encompassed 908120 and 908206 (Table 2).

Under a consistent sowing density of $6.75 \times 10^6$ plants ha$^{-1}$, the parents Jinmai 47 and Jing 411, along with their six NILs, showed varying tillering, leading to different SD. Jinmai 47 and NILs 908206, 808032, and 908274 each displayed higher SD, whereas Jing 411 and NILs 908092, 908120, and 188 each exhibited lower SD, irrespective of water conditions and growing years (Table 3).

Considering both LIA and SD, certain genotypes demonstrated similar LIA but significantly different SD, such as the pair 908274 and Jing 411. Conversely, some had statistically similar SD but different LIA, exemplified by the following five pairs: Jinmai 47 and 908274, Jing 411 and 908120, 908188 and 908206, 908092 and 908120, and 908206 and 908032. These valuable materials enabled a separate analysis of the effects of LIA and SD on WU parameters. Specifically, the effect of LIA on WU could be compared under similar SD, and the impact of SD on WU could be assessed with similar LIA.

## Population WU traits of different wheat genotypes

Genotypic variations in WU parameters (ET, PT, and E) were primarily linked to SD differences. Generally, lower SD genotypes (908092, Jing 411, and 908120) consistently ranked highest in WU parameters ET, PT, and E, whereas higher SD genotypes (908032, Jinmai 47, and 908274) consistently ranked lowest. Genotypes 908188 and 908206 exhibited moderate rankings (Table 4). In the narrower SD range resulting from tillering-related genotypic difference, ET, PT, and E exhibited a declining trend with increasing SD.

The correlation analysis indicated that extremely significant negative correlations existed between ET and SD, with an exception in the 2016–2017 growing season under rain-fed condition ($R = -0.6065$). Conversely, no significant correlation existed between ET and LIA (Table 5). Genotypic comparison further highlighted the superior effect of SD and the negligible effect of LIA on WU (Table 6). Specifically, genotypes with similar LIA and SD exhibited similar ET, while those with similar LIA but differing SDs showed different ETs. Notably, three pairs of genotypes with differing LIA but similar SD each showed comparable ET. An exception was observed in the pair of 908206 and 908032, with similar SD but an 11° LIA difference resulting in different ET. PT showed no correlation with LIA over the three years under two water treatments. In contrast, PT was negatively correlated with SD, and these correlations varied across experimental years. In the driest 2016–2017 growing season, the correlations were not significant. However, in the wettest 2017–2018 growing season, the coefficients reached as high as $-0.8496$ under the rain-fed condition

**Table 3  Shoot density (SD, shoots m⁻²) of different wheat genotypes under rain-fed condition and supplemental irrigation condition.**

| Genotypes | 2015–2016 | 2016–2017 | 2017–2018 |
|---|---|---|---|
| Rain-fed condition | | | |
| 908092 | 627d | 825e | 755cd |
| Jing 411 | 629d | 750f | 710e |
| 908120 | 653cd | 847de | 721de |
| 908188 | 675bcd | 869d | 793bc |
| 908206 | 714ab | 951ab | 823ab |
| 908032 | 735a | 912bc | 803b |
| Jinmai 47 | 692abc | 885cd | 813ab |
| 908274 | 705abc | 967a | 849a |
| Supplemental irrigation condition | | | |
| 908092 | 618d | 771d | 711abc |
| Jing 411 | 631cd | 705e | 645c |
| 908120 | 649bcd | 799cd | 658bc |
| 908188 | 649bcd | 785cd | 674bc |
| 908206 | 669abcd | 826abc | 737ab |
| 908032 | 719a | 868a | 741ab |
| Jinmai 47 | 688abc | 845ab | 778a |
| 908274 | 704ab | 813bcd | 765a |

Notes.
Values followed by same lowercase letter in the same columns are not significantly different at P<0.05 according to the LSD test.

and −0.8602 under the supplemental irrigation condition. It seemed that the effect of SD on PT were more significant when the plentiful rainfall ensured a better growth of wheat crops. However, the coefficients between SD and PT were not higher under the supplemental irrigation condition than under the rain-fed condition.

Compared to PT, E showed closer associations with both LIA and SD. Especially under supplemental irrigation condition, negative correlations existed between SD and E in all three growing seasons and even between LIA and E in two seasons. In the 2015–2016 growing season in particular, the negative correlations between SD and E reached extremely significant levels under different water treatments.

## Significant correlation of LIA with yield

The general trend revealed that Group I genotypes with smaller LIA correlated with lower yields, Group III genotypes with larger LIA had higher yields, and Group II genotypes with medium LIA had moderate yields (Table 7). The correlation analysis confirmed the positive correlation between LIA and yield across the genotypes selected based on LIA (Table 8), indicating an influence of LIA on yield. However, SD did not correlate with yield, indicating that genotypic difference in SD related to tillering did not affect yield at the current sowing density.

**Table 4  Evapotranspiration (ET), transpiration (PT), and soil evaporation (E) for the population of genotypes Jinmai 47 and Jing 411 along with their 6 near isogenic lines (NILs) under rain-fed and supplemental irrigation condition.**

| Genotypes | 2015–2016 | | | 2016–2017 | | | 2017–2018 | | |
|---|---|---|---|---|---|---|---|---|---|
| | ET (mm) | PT (mm) | E (mm) | ET (mm) | PT (mm) | E (mm) | ET (mm) | T (mm) | E (mm) |
| Rain-fed condition | | | | | | | | | |
| 908092 | 58.6a | 42.2a | 16.4c | 63.4a | 46.6a | 16.8b | 64.7a | 45.9a | 18.8cd |
| Jing 411 | 56.4a | 37.5c | 18.9a | 56.3b | 37.3b | 19a | 63.3ab | 42.7b | 20.6a |
| 908120 | 56.4a | 39.1b | 17.3b | 53.9b | 36.6b | 17.3b | 64.4a | 45.5a | 18.9bc |
| 908188 | 49.7b | 33.8d | 15.9cd | 44.5c | 30.4d | 14.1d | 55.7cd | 37.6d | 18.1d |
| 908206 | 47.4b | 32ef | 15.4de | 51.1b | 35.3c | 15.8c | 57.9bc | 38.3c | 19.6b |
| 908032 | 45.7bc | 34.4d | 11.3 h | 42c | 30.6d | 11.4f | 49.3e | 36e | 13.3 g |
| Jinmai 47 | 44bc | 31f | 13fg | 40.9c | 27.8e | 13.1e | 53.7cde | 37d | 16.7e |
| 908274 | 40.5c | 27.9 g | 12.6 g | 44c | 30.2d | 13.8d | 51.4de | 35.8e | 15.6f |
| Mean | 49.8 | 34.7 | 15.1 | 49.6 | 34.4 | 15.2 | 57.6 | 39.9 | 17.7 |
| CV (%) | 6.52 | 4.65 | 7.8 | 5.61 | 4.78 | 7.73 | 5.93 | 5.69 | 7.03 |
| Supplemental irrigation condition | | | | | | | | | |
| 908092 | 72.3ab | 47.7b | 24.6b | 73.9ab | 50b | 23.9b | 72.9ab | 50.4b | 22.5bc |
| Jing 411 | 75.2a | 49.1a | 26.1a | 76.4a | 50.6a | 25.8a | 75.5a | 51.3a | 24.2a |
| 908120 | 70.2ab | 45.8c | 24.4b | 70.7bc | 47.1c | 23.6b | 71.1ab | 48.4c | 22.7b |
| 908188 | 69.5ab | 42.8d | 26.7a | 66.1cd | 43.2e | 22.9c | 67bc | 45.5e | 21.5c |
| 908206 | 66.7bc | 40.7e | 26a | 65.7cd | 40.1 g | 25.6a | 63.6c | 41g | 22.6b |
| 908032 | 61.3c | 45.9c | 15.4e | 64.8de | 46.1d | 18.7d | 64.4c | 46.4d | 18de |
| Jinmai 47 | 52.7d | 36.2f | 16.5d | 60.4e | 42.2f | 18.2d | 61.5c | 44.2f | 17.3e |
| 908274 | 54.4d | 35.9f | 18.5c | 65.2de | 42.4f | 22.8c | 60.7c | 39 h | 21.7bc |
| Mean | 65.3 | 43 | 22.3 | 67.9 | 45.2 | 22.7 | 67.1 | 45.8 | 21 |
| CV (%) | 5.61 | 3.41 | 6.05 | 4.11 | 2.94 | 5.3 | 5.31 | 3.21 | 7.22 |

Notes.

WU for 25 d from 17th April to 11th May in the 2015-2016 growing season; WU for 25 d from 18th April to 12th May in the 2016-2017 growing season; WU for 22 d from 20th April to 11th May for measurement in the 2017–2018 growing season. Values without the same letters in the same column are significantly different at 5% probability level. Values followed by same lowercase letter in the same columns are not significantly different at $P < 0.05$ according to the LSD test.

**Table 5  Coefficients between population transpiration (PT), evaporation (E), and evapotranspiration (ET) with leaf inclination angle (LIA) and shoot density (SD) under rain-fed and supplemental irrigation condition.**

| | LIA | | | SD | | |
|---|---|---|---|---|---|---|
| | 2015–2016 | 2016–2017 | 2017–2018 | 2015–2016 | 2016–2017 | 2017–2018 |
| Rain-fed condition | | | | | | |
| PT | −0.2366 | −0.564 | −0.4949 | −0.7531[*] | −0.5167 | −0.8496[**] |
| E | −0.4772 | −0.6785[*] | −0.4374 | −0.8499[**] | −0.6916[*] | −0.5961 |
| ET | −0.3521 | −0.638 | −0.4615 | −0.8606[**] | −0.6065 | −0.8255[**] |
| Supplemental irrigation condition | | | | | | |
| PT | −0.3357 | −0.4891 | −0.1965 | −0.6029 | −0.6431 | −0.8498[**] |
| E | −0.3635 | −0.7228[*] | −0.7675[*] | −0.8429[**] | −0.7331[*] | −0.6949[*] |
| ET | −0.4074 | −0.7409[*] | −0.4899 | −0.8387[**] | −0.8583[**] | −0.8602[**] |

Notes.

[*,**] Significant relationship at the 0.05 and 0.01 probability level, respectively.
**Table 6  Genotypic comparison in leaf inclination angle (LIA), shoot density (SD), and evapotranspiration (ET) under rain-fed and supplemental irrigation condition.**

| LIA | SD | Genotypes | ET |
|---|---|---|---|
| ≈ | ≈ | 908032 and Jinmai 47 (5° LIA difference) | ≈ |
| ≈ | ≠ | 908274 and Jing 411 (4° LIA difference) | ≠ |
|  |  | Jinmai 47 and 908274 (26° LIA difference) | ≈ |
| ≠ | ≈ | Jing 411 and 908120 (17° LIA difference) | ≈ |
|  |  | 908188 and 908206 (13° LIA difference) | ≈ |

**Table 7  Grain yield (kg ha$^{-1}$) of genotypes Jinmai 47 and Jing 411 along with their six near isogenic lines (NILs) under rain-fed and supplemental irrigation condition.**

| Genotypes | 2015–2016 | 2016–2017 | 2017–2018 |
|---|---|---|---|
| Rain-fed condition |  |  |  |
| Jinmai 47 | 6,597a | 6,277a | 6,430b |
| 908032 | 5,909cd | 5,481cde | 5,968cde |
| 908188 | 5,738de | 6,401a | 6,109bcde |
| 908274 | 5,634de | 5,250ef | 5,719efg |
| 908206 | 5,452efg | 5,584cd | 6,198bcd |
| 908120 | 5,384efg | 4,948g | 5,810def |
| 908092 | 5,058gh | 4,835g | 5,340gh |
| Jing 411 | 4,935 h | 5,085fg | 4,931 h |
| Mean | 5,588 | 5,482 | 5,813 |
| CV (%) | 3.99 | 2.91 | 4.07 |
| Supplemental irrigation condition |  |  |  |
| Jinmai 47 | 8,177a | 7,929a | 8,559a |
| 908032 | 7,631b | 7,974a | 7,802b |
| 908188 | 7,479bc | 7,236bc | 7,341bcd |
| 908274 | 6,901ef | 7,004bcd | 6,446g |
| 908206 | 7,235cd | 6,763def | 7,562bc |
| 908120 | 6,809ef | 6,463ef | 6,890defg |
| 908092 | 6,574f | 6,456f | 6,594fg |
| Jing 411 | 6,601f | 6,657def | 7,122cde |
| Mean | 7,175 | 7,060 | 7,289 |
| CV (%) | 2.37 | 3.58 | 4.05 |

**Notes.**
Values followed by same lowercase letter in the same columns are not significantly different at $P < 0.05$ according to the LSD test.

# DISCUSSION

## A strong negative correlation of SD with ET across different tillering genotypes

WU is influenced by diverse factors encompassing crops, soil and atmosphere. A higher SD implies more shoots utilizing water, resulting in increased ET. However, in this study, SD was negatively correlated with ET ($P > 0.01$) in most trials across three growing seasons under two water treatments. No relevant data on WU correlations with canopy structure

**Table 8** Coefficients between yield with leaf inclination angle (LIA) and shoot density (SD) under rain-fed and supplemental irrigation conditions.

| Experimental year | LIA | | SD | |
|---|---|---|---|---|
| | **Rain-fed** | **Supplemental irrigation** | **Rain-fed** | **Supplemental irrigation** |
| 2015-2016 | 0.8167[**] | 0.8701[**] | 0.6344 | 0.6148 |
| 2016-2017 | 0.7629[*] | 0.7007[*] | 0.2966 | 0.6863[*] |
| 2017-2018 | 0.8608[**] | 0.7273[*] | 0.6603 | 0.3821 |

Notes.
[*],[**] Significant relationship at the 0.05 and 0.01 probability level, respectively.

factors are currently available. The negative correlation was observed between SD and ET across different genotypes rather than within the same genotype across different SDs. This negative correlation between SD and ET was merely found in such narrow SD range caused by variations in tillering ability among genotypes. However, within a larger range, SD might not negatively correlate with ET for the same genotype. Our separate study, investigating the effects of sowing density on WU, incorporated three SDs in a broader range (*Huang et al., 2024*). The findings indicted a general increase in WU with increasing SD (see Table S1). The negative correlation, somehow, suggested that a higher SD does not necessarily result in a greater WU. The underlying mechanism, specific SD range for the negative correlation, and its influencing factors need to be further explored by setting up a more detailed SD gradient.

## Negative correlation of SD with PT implied the predominant effect of canopy structure

SD was negatively correlated with PT, indicating an unexpected decrease in PT with higher SD. This negative correlation implies the presence of an alternative factor influencing PT beyond SD. Rather than transpiration of a single leaf or an individual plant, PT is influenced by canopy structure, creating resistance that regulates air-canopy water and heat exchange. A canopy structure with better permeability facilitates increased transpiration in a population (*Johnson, Witters & Ciha, 1981*). With similar leaf stomatal aperture, genotypic difference in leaf transpiration rate primarily arises from canopy structure-related variations. Thus, the PT genotypic difference is not only affected by leaf stomatal conductance but is strongly affected by canopy structure. SD emerges as a pivotal factor shaping canopy structure, with higher SD resulting in a more complete canopy, augmenting resistance to water and heat dissipation, and consequently suppressing PT. Thus, the negative correlation suggested the substantial impact of canopy structure related to SD.

## Antagonism between the SD-E and SD-T correlations

The higher SD results in a more complete canopy, enhancing efficiency in preventing soil surface evaporation. *Blum (2009)* suggested that a more complete canopy moderated transpiration from the canopy and especially prevented evaporation from the soil surface (*Blum, 2009*). Studies on clump planting of grain sorghum (*Sorghum bicolor* L. Moench)

and maize demonstrated increased canopy closure and reduced WU at the vegetative growth stage (*Kapanigowda et al., 2010*). Consistently, E decreased with increasing SD, and thus negatively correlated with E in the present study.

SD–E and SD–PT correlations varied across growing seasons, displaying apparent antagonism. In the 2017–2018 growing season, SD-PT correlation was higher, while SD-E correlation was lower compared to the other two growing seasons. Although PT and E were essentially antagonistic, they both are affected by soil water conditions and underwent increase under better water availability. Higher rainfall in April 2018 ensured increased wheat crops growth, which certainly prevented E and enhanced PT. If the higher SD–PT correlation in 2018 could be solely attributed to improved crop growth, then under the supplemental irrigation condition, one could expect a higher SD–PT correlation and a lower D–E correlation, assuming better crop growth due to enhanced water availability. However, the study revealed a relatively higher SD–E correlation under supplemental irrigation condition, challenging the expectation. Therefore, the improved crop growth resulting from enhanced water availability could not explain the higher SD–PT and lower SD–E correlations. The observed variation patterns in SD–PT and SD–E correlations across growing seasons became a perplexing phenomenon.

## Strong correlation of LIA with yield for wheat crops

LIA affects canopy foliation arrangement, influencing its light interception. Some previous studies indicated that the upright-leafed genotypes with smaller LIA facilitated deeper penetration and a more uniform distribution of light within the canopy, enhancing photosynthetic efficiency (*Virk, Khush & Peng, 2004*; *Zheng et al., 2008*). In cotton, since the earlier fruits are on the lower parts of the plant, and the ability of older leaves to produce sufficient photosynthates in concert with fruit requirements limits potential yield, adequate light penetration into the crop canopy is particularly important (*Wullschleger & Oosterhuis, 1990*; *Marois et al., 2004*). Contrary findings by *Herbert (1991)* and *Herbert (1992)* indicated that a more erect upper foliage distribution promoted deeper light penetration but reduced light interception by upper layers. Additionally, *Falster & Westoby (2003)* utilized digitizing technology and a three-dimensional architecture model to analyze 38 perennial species only to correlate architectural properties (leaf angle and leaf size) with light capture and simulate carbon gain. They found that shallow-angled leaves had greater total daily light interception and potentially greater day-long carbon gain than steeper-angled leaves.

Differing from the lower plant regions where earlier fruits are found in cotton, the uppermost part in wheat is wheat ear. The yield formation in cereals primarily depends on photosynthates from younger leaves after anthesis, contributing to 70 −80% of grain yield (*Prasad et al., 2008*). In particular, the flag leaf serves as the major photosynthetic apparatus in later developmental stage, with its contribution to grain yield exceeding one-third (*Murchie et al., 1999*). Considering these findings, wheat genotypes with larger LIA should enhance light interception and carbon gain. In our study, LIA positively correlated with yield within an LIA range of 35° through 65° across the genotypes with similar genetic backgrounds (six NILs and their parents), indicating its potential influence on light interception and, consequently, yield. This implied the potential of LIA trait in

wheat yield improvement. However, cautions should be exercised in drawing general conclusions about the impact of LIA on wheat yield, as the results are based on limited genotypes. Therefore, further investigations with a broader range of cultivars featuring diverse LIA values are needed to substantiate these findings.

## CONCLUSIONS

SD was negatively correlated with WU parameters (PT, E, and ET) within a narrow SD range attributed to genotypic tillering difference. However, no correlation with yield was observed. This negative correlation proved the predominant effect of canopy structure on WU, suggesting the potential for water conservation by strategically managing SD to establish an optimal canopy structure. Further exploration is required to clarify the SD range ensuring this negative correlation with WU and its influencing factors. In contrast, LIA exhibited no significant correlation with PT and ET, only correlating with E in specific experimental year. Despite a maximum 26° LIA difference, genotypes with similar SD consumed comparable ET. These findings indicated that intraspecific LIA variation is insufficient to significantly impact canopy structure and, consequently regulate water use in wheat crops. How different LIA might significantly affect canopy structure and WU still remains to be explored. Surprisingly, LIA closely correlated with yield across genotypes with similar genetic backgrounds. This implied that LIA difference influenced light interception, thereby affecting yield. LIA, possibly among the multiple factors affecting yield formation, thus emerges as a potential trait for yield improvement.

## ACKNOWLEDGEMENTS

We heartily thank Professor Ruilian Jing (Institute of Crop Science, Chinese Academy of Agricultural Sciences) for kindly providing the winter wheat NILs.

### Funding

This work was supported by the grant from the National Key R & D Program (2021YFE0101300) and the National Key R & D Program (2023YFD1900503). The funders had no role in study design, data collection and analysis, decision to publish, or preparation of the manuscript.

### Grant Disclosures

The following grant information was disclosed by the authors:
The National Key R & D Program: 2021YFE0101300, 2023YFD1900503.

### Competing Interests

The authors declare there are no competing interests.

## Author Contributions

- Guirong Huang performed the experiments, analyzed the data, prepared figures and/or tables, authored or reviewed drafts of the article, and approved the final draft.
- Xinying Zhang performed the experiments, prepared figures and/or tables, and approved the final draft.
- Zhenzhao Wang performed the experiments, analyzed the data, prepared figures and/or tables, and approved the final draft.
- Xiaoying Liu conceived and designed the experiments, authored or reviewed drafts of the article, and approved the final draft.
- Rui Guo conceived and designed the experiments, authored or reviewed drafts of the article, and approved the final draft.
- Fengxue Gu conceived and designed the experiments, authored or reviewed drafts of the article, and approved the final draft.
- Enke Liu conceived and designed the experiments, authored or reviewed drafts of the article, and approved the final draft.
- Shuying Li performed the experiments, prepared figures and/or tables, and approved the final draft.
- Xiuli Zhong conceived and designed the experiments, analyzed the data, prepared figures and/or tables, authored or reviewed drafts of the article, and approved the final draft.
- Qiaozhen Li conceived and designed the experiments, authored or reviewed drafts of the article, and approved the final draft.
- Xurong Mei conceived and designed the experiments, authored or reviewed drafts of the article, and approved the final draft.

## Data Availability

The raw measurements are available in the Supplementary Files.

## Supplemental Information

Supplemental information for this article can be found online at http://dx.doi.org/10.7717/peerj.17618#supplemental-information.

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
