# Peer review of "Influence of leaf inclination angle and tillering on population transpiration, soil evaporation, and yield in winter wheat near-isogenic lines"

_PeerJ, doi:10.7717/peerj.17618_

## Round 0.1 · original submission · Major Revisions

Dear Authors, according to the reviewers' comments, the manuscript needs a major revision. I highly recommend taking their comments and suggestions into consideration. Specifically, it is suggested to clarify some sentences making them more readable, and to elucidate how the study was conducted and the way in which data was analysed.

Reviewer 1 ·

Basic reporting

The English was relatively clear throughout. However, the article has several spelling errors and instances of awkward grammar (lines 19, 37-38, 52-54, 62-63, 77-78, 99-100, 185-186, etc.). Many sentences had to be re-read multiple times for clarity on authors’ intentions. It took time to decipher which crop was being evaluated (wheat). This should be immediately obvious in the title/abstract.

Line 59: Correctly expressed as denominator and numerator.
Lines 94-98: Clarification - population density is not altered by tillering, as the same number of plants remain. Tillering can mimic a greater plant density, but is not the same as adding more plants.
Lines 103-107: Materials/methods description should not be included in introduction.

Introduction: good order, relatively clear flow of ideas – however, the relationship between leaf angle and tillering was not clear – how/why are these connected? Objectives should be rewritten with a focus on clarity as intentions were unclear.

Experimental design

The research question appears meaningful to this region of China and other such climates globally. Several questions arose on how the project was conducted and specifics of methods.

Line 120: Please provide soil texture information and fertility on a kg ha-1 basis.
Line 121: Were pH tests not done?
Line 121-122: Were soil tests done prior to or following fertilizer applications?
Line 140-141: “Completely level” – is this possible? Is this included to indicate that no runoff was observed in irrigated treatments? In addition – do you believe that a 2.5-m buffer zone was sufficient to prevent any impacts of water movement between plots? 60 mm of water at a time is a significant amount to apply in a single application.
Line 142: Did adding 90 mm irrigation to the entire study negate impacts of water stress on tiller formation at early stages of the growing season (i.e. the reason for multiple water regimes)? Was encouraged tiller formation perhaps a goal? Again, this is a large amount of water.

Note: The use of language “about” and implications of “close enough” is worrisome and appears unscientific. Did authors collect data on these points?

Lines 154-157: Which leaves were considered for LIA measurements? (Line 205: flag leaf?) Were these measurements also taken in central rows? Were all angles averaged from both shoots and tillers? This measurement is questionable as presented.
Line 165: This is the first discussion of a weather station used to collect weather data. What information was gathered and how was this validated?
Line 171: Were no gravimetric soil water content measurements taken prior to irrigation?
Line 191-192: Were border rows excluded? Were tillers/shoots counted again at harvest? Not every shoot will be productive – were only fertile tillers considered or all tillers?

Statistical analysis:
Why were water regimes and years evaluated separately and not included as a random effect for experimental design? (that is, were water conditions truly different? – Table 3 has letters assigned as if they were compared?)
In addition, as multiple factors (with interactions) were considered, why were only 1-way ANOVAs utilized? This appears incorrect and could impact the findings and blanket-significant p-values in the results.

Validity of the findings

Data provided in raw form do not have appropriate headings, dates, or labels to be useful.

Methodological concerns should be discussed and addressed to verify conclusions presented.

Additional comments

This study took relatively intensive field measurements on items that are typically not easy to quantify. Authors are commended for this effort! However, the reporting of how the study was conducted and the way in which data was analyzed has raised some questions.

Reviewer 2 ·

Basic reporting

Many sentences and claims are ambiguous, intelligible and need clarification or rephrasing. Below are some suggestions/questions related to language, and structure that might help understanding the manuscript:

L24 “wheat crops have”
L29 PD wasn’t mentioned previously. Please replace with “population density”
L71 not clear
L74 is spatial distribution of leaves not part of plant architecture?
L88 please replace “plant quantity” with “plant density” as it is easier to understand
L99 where is the spatial distribution of leaves mentioned in L74?
L99 what do you mean by canopy closure? Canopy coverage?
L99 affirmatively? Please use an alternative word.
L100 “thereby conserving”
L101 by canopy closure do you mean canopy coverage?
L105 “The NIL population showed a gradient in mean LIA”
L106 what does "current sowing density" mean?
L110 “The results would provide important information”
L111 “for breeding water saving cultivars”
L131 NCP?
L132 “Both genotypes were backcrossed”
L133 no separation phenomenon? Rephrase.
L203 no need to mention “across two winter wheat genotypes along with their six 204 NILs” in the sections. The reader already knows the material used in this study. You can call sections by traits e.g. LIA and tillering, population water use…
L211 please use groups instead of line names in all manuscript when possible to make it easier to follow. Individual names are confusing and we can’t guess which LIA they have.
L228 remove “To conclude”
L232 what do the symbols mean? Please show statistics with R2.
L233 “were also shown”
L234 “Genotypes with similar LIA”
L245 remove “rather”
L277 remove “incredible”
L280 “The correlation trend between PD and ET might be different if PD range was higher”
L281 unintelligible. Please rephrase.
L286 remove “anyhow”
L292 remove “incredible”
L352 remove “research”

Literature references: it is surprising that there are not more studies mentioned addressing a basic research question such as the effect of plant architecture and density on water use and yield. Some statements need references such as L67-71.

Experimental design

This research investigates the effect of plant architecture and density on water use and yield using wheat lines with similar genetic background, which makes NILs more comparable and conclusions more accurate as it removes genetics as the explanatory factor. It is worth mentioning NILs in the title. Why not state the conclusion of the study in the title?

Methods were described with sufficient detail & information to replicate.

Validity of the findings

The study can be useful to support other studies addressing the same research question.

---

## Round 0.2 · Major Revisions

Dear authors, the MS in this form shows significant improvement, anyway I agree with the reviewers, It still needs further changes especially concerning the statistical session that must be "adjusted" to your experimental design.

Reviewer 1 ·

Basic reporting

English was much improved in this version, with a couple remaining errors.

45-46: "necessitating essential" is redundant
abbreviations should be redefined in the text, not only in the abstract (example: WU, line 56)
86-95: be consistent with sowing/planting density
96: "reduced" - should present tense, not results
107: "affiliated with"
111: remove "determined as"
158-159: "were assumed to be 0" is a much more scientific/professional way to phrase this
183: "from"
217-219: clarify what is meant by "highest" and "lowest" rank ... is this best water use? highest amount of water used?
283: be consistent with corn vs maize

I remain very apprehensive about the use of "population density" in the way this article frames it. Population density is synonymous with planting/sowing density in the sense that it refers to the number of individual plants present. A more accurate and appropriate depiction of what the authors are conveying would be "shoot density" - all plots have the SAME population density in this study - they were planted the same. Tillers are not separate plants but extensions of plants already there. Please change this.

Introduction includes more clear connections. Objectives are better.

Experimental design

Soil information is more suitable in the current form. Please include phrasing in the text that addresses the replies to my previous questions - most concerns were addressed with text additions/revisions, but not all.

Leaf and plot information was more clear in methods. Please include information about the weather station and where it was located.

Remaining concerns:
If water regimes were separate experiments, they should not be described as "treatments" - instead they could be described as "environments"?
I maintain that a one-way anova is not appropriate here. You have two levels of water you are considering in these models and eight levels of genotype, including their interactions. Please run a two-way anova, adjust the structure of your model, or provide more significant rationale for why you are doing it this way.

Validity of the findings

Of concern based on the statistical methods utilized. Please re-assess.

Related to discussion, if the data referenced in 259-264 is published elsewhere, it needs to be cited. Otherwise, there is not any information on how this study was conducted. What were the treatments? How was it measured? Were similar genotypes used? Were similar environments evaluated?

Additional comments

This revision was a significant improvement over the first version, but the manuscript requires additional work. Reassessing the analysis is strongly suggested.

Reviewer 2 ·

Basic reporting

The English is much clearer and comments were addressed.

Experimental design

Comments were addressed.

Validity of the findings

Comments were addressed.

Additional comments

Minor comments:

L60 “over 70% of water irrigation”

L77 by E you mean ET?

L103 needs reference

L112 This sentence “The findings offer valuable insights for breeding water-saving cultivars based on plant architecture traits and optimizing population structure for effective field water conservation.) better fits with Discussion or Conclusion sections.

---

## Round 0.3 · Minor Revisions

In agreement with one of the reviewers, the authors should think to use the term “shoot density” according also to the definition previously gave in the rebuttal letter, I agree that it is more appropriate.

In M&M section the authors stated that: “A one-way ANOVA (Analysis of variance) was carried out in experimental indices (PD, three WU traits, and yield) under each water condition per growing season to primarily evaluate the differences across eight genotypes.” As you describe here, it seems that a two-way ANOVA could be more appropriate. But I read your rebuttal letter, in which you better explain the experimental design. So please, modify the M&M integrating it with the the explanation given in the Rebuttal letter in which you specify why the one-way ANOVA was used instead of two-way ANOVA.

Reviewer 1 ·

Basic reporting

Authors adjusted English and some references as suggested but are not willing to change the term "population density" to something more appropriate - although they defined it as shoot density (so why not use "shoot density"?). This terminology remains very misleading.

Experimental design

The design of the experiment with relation to the water treatments and how they were handled in the analysis remains arguably inappropriate. One-way anovas are not meant to handle interrelationships of two independent variables, as presented in the tables here. Authors pasted the same response as before without attempting suggested revisions.

Validity of the findings

Remain of questionable quality.

Additional comments

I am not convinced of your position(s) and maintain mine.

Reviewer 2 ·

Basic reporting

NA

Experimental design

NA

Validity of the findings

NA

---

## Round 0.4 · accepted · Accept

The authors have addressed all of the comments. I revised the final version and I am satisfied of the current version, hence I believe that the manuscript ready for publication. Regards FC